# Can Amygdala-Derived-EEG-fMRI-Pattern (EFP) Neurofeedback Treat Sleep Disturbances in PTSD?

**DOI:** 10.3390/brainsci15030297

**Published:** 2025-03-12

**Authors:** Aron Tendler, Yaki Stern, Tal Harmelech

**Affiliations:** Gray Matters Health, Haifa 3303403, Israel; yaki@graymatters-health.com (Y.S.); talharmelech@gmail.com (T.H.)

**Keywords:** amygdala-EFP neurofeedback, self-neuromodulation, emotional regulation, PTSD, comorbid insomnia, nightmares, symptoms clusters, temporal dynamics of symptoms improvement

## Abstract

Background: Sleep disturbances are a core feature of post-traumatic stress disorder (PTSD), affecting up to 90% of patients and often persisting after standard PTSD treatment. As all the current interventions have limitations, amygdala-targeted neuromodulation may offer a novel treatment pathway. Methods: Secondary analysis of a prospective, single-arm trial (n = 58) was carried out evaluating Prism™ amygdala-derived-EEG-fMRI-Pattern neurofeedback (Amyg-EFP-NF). Sleep outcomes were assessed using the Clinician-Administered PTSD Scale (CAPS-5) sleep item, PTSD Checklist (PCL-5) sleep item, and Patient Health Questionnaire (PHQ-9) sleep items at baseline, post-treatment, and 3-month follow-up. Treatment consisted of 15 sessions over 8 weeks. Results: At 3-months’ follow-up, 63.79% of participants demonstrated clinically meaningful reduction in sleep disturbances (≥1 point reduction in CAPS-5 Item 20). Sleep improvement showed a moderate correlation with overall PTSD symptom reduction (r = 0.484, *p* < 0.001) and a balanced improvement pattern (−15.1% early, −9.1% late). Sleep responders sustained improvement across multiple measures and showed significant increases in cognitive reappraisal (mean change: +2.57 ± 1.0, *p* = 0.006), while non-responders showed initial but un-sustained improvement in trauma-related dreams. Conclusions: Amyg-EFP-NF shows preliminary promise for treating PTSD-related sleep disturbances. Our exploratory analyses suggest distinct temporal patterns of sleep improvement and potential associations with enhanced cognitive reappraisal capacity that warrant rigorous investigation in future randomized controlled trials.

## 1. Introduction

Sleep disturbances represent a significant challenge in post-traumatic stress disorder (PTSD), with high prevalence rates and substantial impact on patient functioning [1,2,3]. These disturbances, including difficulties in initiating and maintaining sleep, contribute substantially to daytime dysfunction and overall disease burden, meeting diagnostic criteria specified in DSM-5-TR [4]. Sleep disruption in PTSD presents as a complex phenomenon, often manifesting as both a core symptom of the disorder and an independent maintenance factor that can perpetuate other PTSD symptoms [1,2,5]. The persistence of sleep problems even after successful PTSD treatment highlights the need to understand their unique recovery mechanisms.

The complex relationship between sleep disturbance and emotional dysregulation forms a potentially self-reinforcing cycle central to PTSD maintenance [6]. Insufficient sleep amplifies emotional reactivity and decreases emotion regulation capacity through altered amygdala function and reduced medial prefrontal cortex–amygdala functional connectivity [7,8,9]. This heightened emotional reactivity may, in turn, further compromise sleep quality, creating a perpetuating cycle [8]. The amygdala plays a pivotal role in this relationship, showing heightened reactivity during sleep deprivation and contributing to both emotional processing difficulties and sleep disruption [7].

The temporal dynamics of symptom improvement in PTSD treatment have garnered increasing attention, particularly regarding sleep disturbances. While some PTSD symptoms may show rapid response to intervention [1], sleep disturbances often demonstrate distinct recovery patterns, potentially reflecting unique underlying mechanisms. Understanding these temporal patterns and their relationship with emotional processing changes could provide crucial insights into treatment mechanisms, particularly given the demonstrated capacity for volitional neuromodulation to influence both emotional regulation and sleep quality [10].

Current treatment options for sleep disturbances in PTSD have significant limitations. While cognitive behavioral therapy for insomnia (CBT-I) shows efficacy [11], access is restricted by provider availability and specialized training requirements [1,12]. Pharmacological approaches carry risks of side effects, dependence, and potential interference with emotional processing and fear extinction [4]. Moreover, evidence suggests that sleep disturbances with short sleep duration may be less responsive to standard protocols [7], highlighting the need for novel treatment approaches that can address both sleep and emotional regulation components of PTSD.

The emergence of neuromodulation techniques offers promising new avenues for treating both PTSD and sleep disorders [2], with evidence supporting the modulation of amygdala activity and arousal through neurofeedback (NF) approaches [13]. Recent advances in our understanding of neural circuitry have highlighted the role of amygdala hyperactivity in both conditions, suggesting potential therapeutic benefits from interventions targeting this mechanism [7,8]. The amygdala-derived-EEG-fMRI-Pattern neurofeedback (Amyg-EFP-NF) system represents a novel approach that enables patients to self-regulate amygdala activity through implicit emotional regulation training [14]. The Amyg-EFP-NF system represents an innovative approach to targeted neuromodulation. This technology integrates simultaneous EEG and fMRI recordings through a machine learning algorithm that identifies specific EEG patterns correlating with amygdala activity [15]. Unlike traditional NF methods that have low spatial resolution and cannot capture neural activity from deeper brain regions, the Amyg-EFP approach enables indirect access to amygdala functioning using only scalp EEG. The Prism™ system operationalizes this technology by providing real-time feedback based on these amygdala-derived EEG patterns through an audio–visual interface. This allows patients to modulate their emotional state implicitly, without requiring explicit strategy instructions, potentially facilitating downregulation of the amygdala hyperactivity associated with both PTSD symptoms and sleep disturbances [14,16].

Emotion regulation strategies, particularly cognitive reappraisal, may play crucial roles in sleep disturbance maintenance and recovery [8,17]. Cognitive reappraisal involves reinterpreting emotional stimuli to modify their impact and has been associated with better psychological outcomes and treatment response across various disorders [18]. Understanding how these strategies change during treatment could illuminate mechanisms linking emotional processing to sleep improvement. Advances in understanding sleep–emotion interactions, combined with new capabilities for targeting amygdala activity, suggest potential therapeutic approaches for addressing sleep disturbances in PTSD.

The current study examined whether successful amygdala regulation would lead to improved sleep outcomes in PTSD patients. We focused particularly on temporal patterns of improvement and the role of emotion regulation strategies, hypothesizing that (1) sleep improvement patterns would differ from other PTSD symptoms, reflecting distinct recovery mechanisms, and (2) successful sleep improvement would be associated with enhanced emotional regulation capacity. Understanding these relationships could inform more targeted and effective interventions for sleep disturbances in PTSD.

## 2. Methods

### 2.1. Study Design and Participants

This secondary analysis utilized data from a prospective, single-arm, open-label trial evaluating Prism™ as adjunct treatment in chronic PTSD [14]. Eligible participants were adults aged 22–65 years with DSM-5 PTSD and index trauma 1–20 years prior (Table 1), and baseline sleep disturbance (CAPS-5 Item 20 > 0). Power analysis indicated the sample size (n = 58) provided adequate power (80%) to detect moderate effect sizes (d = 0.5) in sleep in primary sleep outcomes and 70% power for secondary outcomes. Missing data were minimal (<5%) due to a high retention rate. Complete case analysis was used for our primary analyses, as the small amount of missing data were determined to be missing completely at random based on the distribution of missing values. Recruitment occurred through clinical referrals and advertisements at five clinical centers: four in Israel (Rambam Medical Center, Sheba Medical Center, Be’er Ya’akov Mental Health Center, Barzilai Medical Center), and one in the United States (New York University Langone Health). The study was approved by the ethics committees at each participating clinical site and the trial was registered on ClinicalTrials.gov (NCT04891614).

### 2.2. Measures

#### 2.2.1. Sleep Assessment and Validation

Sleep disturbances were evaluated using multiple complementary measures: (1) the Clinician-Administered PTSD Scale for DSM-5 (CAPS-5)-Item 20, providing clinician-rated assessment of sleep disturbance on a 0–4 scale; (2) the PTSD Checklist for DSM-5 (PCL-5) Item 20, providing corresponding self-reported sleep assessment on the same 0–4 scale; and (3) PHQ-9 Items 3 (trouble falling/staying asleep) and 4 (fatigue), assessing sleep quality and related daytime dysfunction on a 0–3 scale. The CAPS-5 and PCL-5 sleep items have demonstrated strong psychometric properties and are considered gold standards for assessing PTSD-related sleep disturbance [19]. PHQ-9 sleep items show excellent correlation with the Insomnia Severity Index in both primary care (r = 0.75, *p* < 0.0001) [20] and specialized clinical populations (r = 0.72, *p* < 0.001) [21]. Sleep response was defined as ≥1 point reduction in CAPS-5 Item 20 at 3-month follow-up, representing a clinically meaningful change from a higher to lower symptom intensity category [19]. Changes were analyzed across early (baseline to post-treatment) and late (baseline to follow-up) phases using within- and between-group *t*-tests.

#### 2.2.2. Emotion Regulation Assessment

Emotion regulation strategies were assessed using the Emotion Regulation Questionnaire (ERQ [22]), a validated measure comprising two subscales: cognitive reappraisal (CR) and expressive suppression (ES). The CR subscale (6 items) measures the tendency to reappraise emotional experiences, while the ES subscale (4 items) assesses the tendency to suppress emotional expressions. The ERQ uses a 7-point Likert scale (1 = strongly disagree to 7 = strongly agree). Scores on the CR subscale range from 6–42, scores on the ES subscale range from 4–28, with higher scores indicating greater use of each strategy. The ERQ has demonstrated good internal consistency (α = 0.79 for CR, α = 0.73 for ES) and test-retest reliability (r = 0.69 for both subscales [22]).

#### 2.2.3. PTSD Symptom Assessment

PTSD symptoms were assessed using the CAPS-5, providing severity scores for four symptom clusters: re-experiencing (Criterion B: 5 items, including recurrent distressing dreams), avoidance (Criterion C: 2 items), negative alterations in cognitions and mood (Criterion D: 7 items), and arousal (Criterion E: 6 items, including sleep). The CAPS-5 uses a 5-point severity scale (0 = absent to 4 = extreme/incapacitating) for each symptom. The total CAPS-5 score ranges from 0–80, with scores categorized as: minimal (0–15), mild (16–25), moderate (26–45), severe (46–65), and extreme (66–80). For cluster analyses, the arousal cluster was calculated excluding the sleep item (E6) to avoid redundancy.

### 2.3. Intervention

Participants completed 15 Prism™ NF sessions over 8 weeks. Sessions occurred twice weekly on non-consecutive days, each lasting approximately 30 min with rest periods between trials. Each session provided real-time feedback based on amygdala-derived EEG patterns displayed as an audio–visual interface, enabling implicit emotion regulation training [14]. Participants were instructed to regulate their emotional state using the feedback without explicit strategy instructions, allowing for individualized learning.

### 2.4. Statistical Analysis

Changes in emotion regulation strategies were compared between sleep responders and non-responders using independent *t*-tests. Chi-square tests were used to compare categorical variables, including treatment response rates between trauma types. Pearson correlation coefficients were calculated to examine relationships between sleep measures and between sleep improvement and overall PTSD symptom reduction. To examine temporal patterns, we defined early improvement as change from baseline to post-treatment (8 weeks), and late improvement as change from post-treatment to 3-months’ follow-up. For each symptom cluster, we calculated percentage improvement in each phase by dividing the absolute change by the maximum possible score for that cluster, allowing for standardized comparison across clusters. The late/early improvement ratio was calculated by dividing the percentage improvement in the late phase by the percentage improvement in the early phase. A ratio of 1 would indicate equal improvement in both phases, while values below 1 indicate greater early improvement, and values above 1 indicate greater late improvement. Between-cluster differences in these ratios were tested using Fisher’s z-transformation with independent *t*-tests, using sleep as the reference cluster, as sleep improvement was our primary outcome of interest.

## 3. Results

### 3.1. Overall Treatment Response

Of 63 enrolled participants, 58 completed all assessments through 3-month follow-up (92.1% retention). Dropouts did not differ significantly from completers in baseline characteristics (all *p* > 0.2). Treatment response did not differ significantly between military (64.3%) and civilian (63.2%) trauma (χ^2^ = 0.008, *p* = 0.927), suggesting broad applicability across trauma types.

At 3-months’ follow-up, 63.79% of participants demonstrated a clinically meaningful reduction in sleep disturbances (≥1 point reduction in CAPS-5 Item 20). Effect sizes were large for responders (Cohen’s d = 0.82, 95% CI [0.56, 1.08]) and moderate for the full sample (d = 0.54, 95% CI [0.31, 0.77]), indicating robust treatment effects. Sleep improvement showed moderate correlation with overall PTSD symptom reduction (r = 0.484, *p* < 0.001; adjusted CAPS-5 excluding Item 20: r = 0.405, *p* < 0.005), suggesting partial independence of sleep recovery mechanisms from overall symptom improvement (Figure 1).

### 3.2. Measurement Validation and Convergence

The validity of sleep improvement measurement was supported by strengthening correlations between assessment methods over time (Figure 2A, Table 2). The association between clinician-rated (CAPS-5 Q20) and self-reported (PCL-5 Q20) sleep disturbance showed a significant increase from baseline (r = 0.49) to follow-up (r = 0.68, *p* < 0.01). Sleep disturbance measured by PHQ-9 Q3 showed moderate but non-significantly increasing correlations with CAPS-5 Q20 (r = 0.35 to 0.6), while fatigue (PHQ-9 Q4) showed consistently weaker associations (r = 0.11 to 0.32).

### 3.3. Differential Patterns in Sleep-Related Symptoms

Analysis of specific sleep-related symptoms revealed distinct patterns between responders and non-responders (Table 3, Figure 2B). While clinician-rated sleep improvement (CAPS-5 Q20) showed expected differences given the response definition, self-reported measures provided independent validation of these patterns. Self-reported sleep disturbance (PCL-5 Q20) showed progressive improvement in responders (early: −0.57 ± 0.12, *p* = 0.005; late: −1.14 ± 0.22, *p* < 0.001) but no significant change in non-responders, with increasing differentiation between groups (early: *p* = 0.04; late: *p* < 0.001). Sleep initiation/maintenance problems (PHQ-9 Q3) showed an even more pronounced pattern, with responders demonstrating substantial late-phase improvement (−3.9 ± 0.26, *p* < 0.001) while non-responders showed no significant change. Notably, trauma-related dreams (CAPS-5 Q2) showed a unique pattern: both groups demonstrated significant early improvement (responders: −0.43 ± 0.23, *p* = 0.031; non-responders: −0.57 ± 0.27, *p* = 0.02), but only responders maintained this improvement at follow-up (responders: −0.46 ± 0.16, *p* = 0.002; non-responders: 0.05 ± 0.31, *p* = 0.44). This pattern of sustained improvement in responders across multiple sleep-related symptoms suggests that successful sleep recovery involves mechanisms that extend beyond initial symptom reduction.

### 3.4. Patterns of Improvement

Sleep improvement demonstrated a distinct temporal pattern compared to other symptom clusters, with late/early improvement ratios revealing notable differences between clusters (Figure 3A). Sleep improvement showed a balanced pattern across phases (−15.1% early phase, −9.1% late phase) with a late/early ratio of 0.60, contrasting with other symptom clusters that showed predominantly early improvement followed by minimal late-phase change: arousal (−11.0% early, −4.1% late, ratio 0.38), cognitions/mood (−18.7% early, +2.2% late, ratio −0.12), avoidance (−12.7% early, −3.7% late, ratio 0.29), and re-experiencing (−15.0% early, −2.4% late, ratio 0.16).

When examining total symptom reduction across the study period, overall improvement was greatest for sleep (−38.5%, SEM = 7%) followed by re-experiencing (−30.7%, SEM = 6%), cognitions/mood (−28.8%, SEM = 4%), arousal (−24%, SEM = 7%), and avoidance (−21.7%, SEM = 7%). Comparative statistical analysis revealed highly significant differences between the late/early improvement ratio for sleep and that of cognitions/mood (*p* = 0.0001), significant differences with re-experiencing (*p* = 0.0094), a marginally significant difference with avoidance (*p* = 0.0541), and no significant difference with arousal (*p* = 0.1383). While other symptom clusters showed predominantly early improvement followed by minimal late-phase change or slight regression, sleep disturbances exhibited a more nuanced recovery pattern with substantial improvement continuing into the late phase.

We examined correlations between sleep improvement and changes in specific PTSD symptom clusters (Figure 3B). Sleep improvement (change in CAPS-5 Item 20) showed the strongest correlation with improvement in the arousal cluster (excluding the sleep item) (r = 0.46, *p* < 0.001), followed by avoidance symptoms (r = 0.33, *p* = 0.012) and re-experiencing symptoms (r = 0.29, *p* = 0.027). The correlation with negative alterations in cognition and mood was weaker and did not reach statistical significance (r = 0.19, *p* = 0.153). The stronger association with other arousal symptoms suggests potential shared regulatory mechanisms, consistent with neurobiological models of hyperarousal affecting both daytime symptoms and sleep disruption.

### 3.5. Treatment-Related Changes in Emotion Regulation and Re-Experiencing Symptoms

Analysis of emotion regulation strategies revealed that sleep responders (n = 37) showed significant increases in cognitive reappraisal from baseline to follow-up (mean change: +2.57 ± 1.0, *p* = 0.006, Figure 4), while non-responders (n = 21) showed no significant changes (+2.76 ± 2.34, NS). Neither group showed significant changes in expressive suppression (responders: −0.68 ± 0.98, NS; non-responders: +0.14 ± 1.01, NS). Examination of trauma-related dreams (CAPS-5 Item 2) revealed parallel patterns: while both groups showed significant early reduction in distressing dreams (responders: −0.43 ± 0.23, *p* < 0.05; non-responders: −0.57 ± 0.27, *p* < 0.05), only sleep responders maintained this improvement at 3-month follow-up (responders: −0.46 ± 0.16, *p* < 0.005; non-responders: 0.05 ± 0.31, *p* = 0.44). This pattern of sustained improvement in both sleep quality and dream content among responders, coupled with their enhanced cognitive reappraisal capacity, suggests a potential link between emotional regulation and maintenance of sleep-related therapeutic gains.

### 3.6. Predictors of Treatment Response

We examined baseline characteristics of sleep responders (n = 37) versus non-responders (n = 21) to identify potential predictors of treatment response. There were no significant baseline differences in demographic factors including age (responders: 38.95 ± 10.42 years, non-responders: 38.33 ± 10.58 years, *p* = 0.83), gender distribution (responders: 46% female, non-responders: 48% female, *p* = 0.90), or trauma type (responders: 51% non-military, non-responders: 48% non-military, *p* = 0.85).

Clinical characteristics including time from symptom onset (responders: 8.35 ± 0.89 years, non-responders: 10.91 ± 1.38 years, *p* = 0.11), baseline PTSD severity (CAPS-5 total: responders 43.19 ± 2.00, non-responders 42.71 ± 2.35, *p* = 0.87), and symptom cluster severity (all *p* > 0.30) also did not significantly differ between groups. Baseline emotion regulation capacity, as measured by the ERQ cognitive reappraisal (responders: 24.35 ± 1.48, non-responders: 25.00 ± 1.93, *p* = 0.78) and expressive suppression (responders: 15.43 ± 0.97, non-responders: 15.29 ± 1.41, *p* = 0.93) subscales, was similar in both groups.

However, baseline sleep disturbance severity (CAPS-5 Item 20) was marginally higher in responders (2.70 ± 0.16) compared to non-responders (2.29 ± 0.18, *p* = 0.08), suggesting that patients with more severe sleep problems may have greater potential for improvement. This finding aligns with our categorical analysis of response rates by baseline severity, which showed increasing response rates with higher baseline sleep disturbance severity.

Further analysis of response rates by baseline sleep severity revealed that among participants with mild baseline sleep disturbance (CAPS-5 Item 20 = 1, n = 8), 50% responded at 3-month follow-up, compared to 61% of those with moderate severity (Item 20 = 2, n = 18), 58% with severe disturbance (Item 20 = 3, n = 24), and 100% of those with extreme severity (Item 20 = 4, n = 8). Similarly, higher overall PTSD symptom severity predicted better sleep outcomes, with response rates of 58.6% for moderate PTSD (baseline CAPS-5 total: 21–40, n = 29), 63.0% for severe PTSD (CAPS-5 total: 41–60, n = 27), and 100% for the most severe cases (CAPS-5 total: 61–80, n = 3) at 3-month follow-up.

These findings suggest that Amyg-EFP-NF may be particularly beneficial for patients with more severe sleep disturbances, possibly due to the intervention’s targeted impact on limbic hyperarousal mechanisms that may be more pronounced in severe cases.

## 4. Discussion

This study provides preliminary evidence regarding PTSD-related sleep disturbance treatment through two noteworthy findings that warrant further investigation. First, sleep improvement showed a unique temporal pattern characterized by substantial late-phase changes, contrasting with the predominantly early improvement seen in other PTSD symptoms. Second, sleep improvement was associated with enhanced cognitive reappraisal capacity, suggesting a potential role for emotional regulation in sleep recovery. While these findings need validation in controlled trials, they provide novel insights into potential mechanisms of sleep improvement in PTSD treatment.

Exploratory analysis of temporal patterns revealed a distinctive characteristic of sleep recovery. The late/early improvement ratio for sleep (0.60) demonstrated a more balanced trajectory compared to other symptom clusters, suggesting a sustained and consistent improvement process. This contrasts with the front-loaded improvements observed in other symptom domains. Statistically significant differences between the sleep cluster and other clusters, particularly cognitions/mood (*p* = 0.0001) and re-experiencing (*p* = 0.0094), provide preliminary evidence of unique recovery mechanisms. This sustained nature of sleep improvement might indicate ongoing reciprocal benefits between improved emotion regulation and sleep quality, aligning with neurobiological models of sleep–emotion interactions [23]. While most PTSD symptoms showed predominantly early improvement with subsequent plateauing or slight regression, sleep disturbances exhibited a more nuanced recovery pattern. This suggests that sleep recovery may involve different mechanisms requiring more time for neural and behavioral adaptation, potentially reflecting the complex interplay between emotional processing and sleep regulation [7,8].

Pace-Schott and colleagues’ recent review offers a potential mechanistic explanation for our observed balanced improvement pattern. They emphasize sleep’s critical role in emotional memory processing, particularly in integrating and reprocessing traumatic memories. Our findings of sustained sleep improvement, coupled with enhanced cognitive reappraisal capacity, align with emerging models suggesting that sleep supports more complex emotional memory reorganization beyond simple symptom reduction [23,24].

These patterns should be interpreted within the limitations of our single-arm design. However, they provide valuable insights into the unique aspects of sleep recovery in PTSD treatment and highlight the need for further research in this area.

The differential patterns of improvement across sleep measures, particularly in trauma-related dreams, provide key mechanistic insights. While both responders and non-responders showed an initial reduction in distressing dreams, only responders maintained this improvement at follow-up, paralleling findings on nightmare recurrence patterns in PTSD treatment [25]. The strengthening correlation between clinician-rated and self-reported sleep disturbance (r = 0.49 to 0.68, *p* < 0.01) further suggests progressive alignment between objective and subjective sleep improvements. This convergence of findings—maintained improvement in trauma-related dreams, enhanced cognitive reappraisal capacity, and strengthening assessment correlations—supports theories that effective processing of trauma-related sleep content requires robust emotion regulation capabilities [8,26]. The pattern also aligns with recent neurobiological models suggesting that improved prefrontal-amygdala regulation may facilitate both better emotional processing and more restful sleep [7].

The differential correlations between sleep improvement and various PTSD symptom clusters provide further evidence for the interconnected but distinct mechanisms of recovery. The strongest correlation with the arousal cluster (r = 0.46, *p* < 0.001) aligns with theoretical models suggesting shared neurobiological pathways between sleep disturbance and hyperarousal symptoms, both potentially mediated by similar limbic system dysregulation. The weaker association with negative alterations in cognition and mood suggests that these symptom domains may involve more distinct recovery mechanisms. These findings support the potential specificity of amygdala-targeted interventions for symptoms with stronger limbic system involvement, particularly sleep and arousal symptoms, and highlight the importance of considering symptom cluster profiles when evaluating treatment efficacy.

The observed associations between cognitive reappraisal and sleep improvement provide intriguing preliminary support for emotional regulation as a potential recovery mechanism. Sleep responders showed significant increases in cognitive reappraisal, possibly reflecting an enhanced capacity for processing emotion-laden memories that might otherwise disrupt sleep. While the causal relationship remains to be established in controlled studies, these findings align with the theoretical model proposed by Ben Simon et al. (2020) [7] regarding the bidirectional relationship between sleep and emotional processing. The pattern suggests that improved emotional regulation may help break the cycle of sleep disruption, though this hypothesis requires further validation. These observations suggest potential mechanistic differences from conventional PTSD sleep treatments.

Unlike traditional sleep interventions in PTSD which typically target sleep directly through behavioral modifications or pharmacology, Amyg-EFP-NF’s effects on sleep may be mediated through enhanced emotional regulation capacity. This proposed mechanism could be particularly relevant for PTSD-related sleep disturbances, where emotional dysregulation plays a central role in sleep disruption. Future controlled studies should examine this potential mechanistic pathway.

While our findings suggest a potential link between improved emotional regulation and sleep outcomes, we acknowledge the absence of direct neurobiological data to establish this mechanistic relationship. Our observed associations between cognitive reappraisal and sleep improvement provide intriguing but preliminary support for emotional regulation as a potential recovery mechanism. Future studies should incorporate neuroimaging measures before and after treatment to directly assess changes in prefrontal-amygdala connectivity and their relationship to both emotional regulation capacity and sleep quality improvements.

Our exploratory analyses suggest that the link between amygdala regulation and sleep improvement may operate through multiple pathways that warrant further investigation. First, reduced amygdala reactivity could lower physiological arousal during the pre-sleep period, facilitating sleep onset. Second, improved emotion regulation capacity may help process trauma-related content that typically disrupts sleep maintenance. Third, the implicit learning nature of the NF protocol may create lasting changes in emotional processing that persist during sleep, potentially affecting both sleep architecture and dream content. These hypothesized mechanisms should be tested in future studies incorporating objective sleep measures and detailed examination of neural activity patterns.

## 5. Clinical Implications

These preliminary findings suggest several considerations for implementing and studying Amyg-EFP-NF in clinical settings. The observed gradual improvement pattern suggests that longer treatment protocols may be beneficial; future research should investigate whether extended treatment courses (e.g., 12–15 weeks) enhance outcomes for sleep-focused interventions. The association between cognitive reappraisal and sleep improvement suggests the potential value of studying combined approaches incorporating explicit emotion regulation training. While the treatment showed similar effectiveness across trauma types in our sample, suggesting potential broad applicability, controlled trials are needed to identify optimal protocols for specific populations and to establish definitive efficacy. When situating these findings within the broader treatment landscape for PTSD-related sleep disturbances, it is important to consider how Amyg-EFP-NF compares with established interventions.

Our findings should be considered in the context of existing evidence-based PTSD sleep treatments. Cognitive behavioral therapy for insomnia (CBT-I) is well established as the first-line treatment for insomnia in PTSD, with meta-analyses reporting moderate to large effects [11]. The effect size observed in our study (d = 0.54 for the full sample) is comparable to those reported for sleep treatments in PTSD populations. Recent network meta-analysis by Huang et al. [2] identified CBT-I as having the highest efficacy for improving sleep quality (SMD = −5.61, 95%CI: −8.82 to −2.40 compared to placebo), while pharmacological treatments showed varying degrees of effectiveness. For nightmare reduction, this same analysis found prazosin (SMD = −1.20, 95%CI: −1.72 to −0.67) and imagery rehearsal therapy (IRT) (SMD = −0.65, 95%CI: −1.00 to −0.31) to be most effective. Our observed effects on trauma-related dreams, particularly the sustained improvement in responders, suggest potential efficacy in this domain as well, though direct comparative studies are needed. The Amyg-EFP-NF approach differs mechanistically by targeting emotional regulation circuits rather than directly modifying sleep behaviors (as in CBT-I) or pharmacologically altering adrenergic activity (as with prazosin). This mechanistic difference may offer advantages for patients who have not responded to conventional approaches or prefer non-pharmacological treatments. Additionally, our finding that patients with more severe baseline sleep disturbances showed better responses contrasts with evidence suggesting that traditional approaches may be less effective for severe cases, particularly those with short sleep duration [3].

## 6. Limitations and Future Directions

Several important limitations must be considered when interpreting our findings. First and foremost, the single-arm design without a control condition precludes definitive conclusions about treatment efficacy, as improvements could reflect natural symptoms fluctuations, regression to the mean, or non-specific treatment effects. While the blinding of assessment timepoints during clinician ratings helps mitigate some potential biases, the exploratory nature of this analysis means our findings should be considered to be hypothesis generating rather than confirmatory. The results presented here provide preliminary evidence that justifies further investigation into properly controlled trials.

Additionally, while our sleep measures showed good convergence, the absence of objective sleep measurements (e.g., actigraphy or polysomnography) limits our ability to assess specific sleep architecture changes. Future studies should incorporate these objective measures to better characterize the nature of sleep improvements and potential mechanisms. These objective measures could help differentiate effects on different sleep parameters (e.g., sleep onset latency, sleep efficiency, REM architecture) and provide insights into potential relationships between objectively measured sleep quality and subjective improvements.

Our study assessed outcomes only up to three months post-treatment, which may not capture the full trajectory of sleep improvement or potential relapse patterns. Longer follow-up periods (e.g., 6–12 months) in future studies would provide important information about the durability of treatment effects and could help identify factors associated with sustained improvement versus symptom recurrence.

Future research should employ randomized controlled designs with comprehensive sleep assessment batteries, including objective measures. Investigation of potential mediating factors between emotion regulation changes and sleep improvement could further elucidate recovery mechanisms. Additionally, examination of individual differences in temporal patterns of improvement could help identify optimal treatment durations.

## 7. Conclusions

Amyg-EFP-NF shows promise for treating PTSD-related sleep disturbances. The distinct temporal pattern of sleep improvement and its association with emotional regulation changes suggest unique recovery mechanisms. These preliminary findings suggest the potential value of targeted interventions that account for the temporal dynamics of sleep recovery, though further investigation in controlled trials is needed to establish optimal treatment protocols.

## 8. Statement of Significance

Sleep disturbances are a core feature of post-traumatic stress disorder (PTSD) that often persist despite treatment. This study provides the first evidence that amygdala-derived-EEG-fMRI-Pattern neurofeedback may improve sleep disturbances in PTSD patients through emotional regulation mechanisms. These findings suggest a novel therapeutic approach for addressing sleep problems in PTSD populations.

## Figures and Tables

**Figure 1 brainsci-15-00297-f001:**
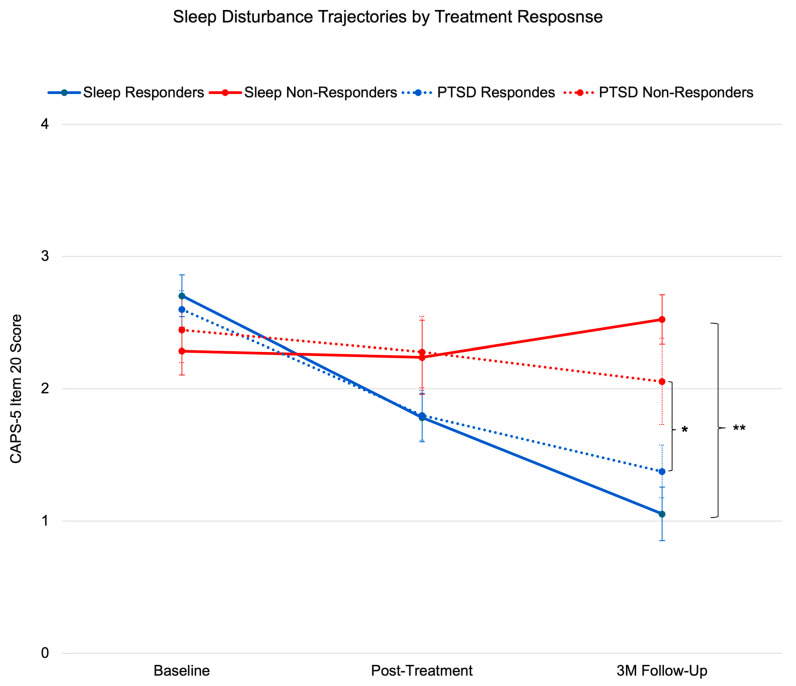
Sleep disturbance trajectories by treatment response. Mean CAPS-5 Item 20 scores (±SE) in treatment responders (sleep responders ≥1-point reduction in CAPS-5 Q20 score, n = 37; PTSD responders ≥6-point reduction in adjusted CAPS-5 score, n = 40) and non-responders (sleep: n = 21; PTSD: n = 18) across timepoints. Despite similar baseline scores (*p* > 0.05), responders showed significantly greater improvement at 3-month follow-up (d = 0.54, *p* < 0.001). Note: Response defined as ≥6-point reduction in adjusted CAPS-5 score (excluding Item 20) to avoid circularity. Error bars represent standard error of the mean. * *p* < 0.01, ** *p* < 0.001.

**Figure 2 brainsci-15-00297-f002:**
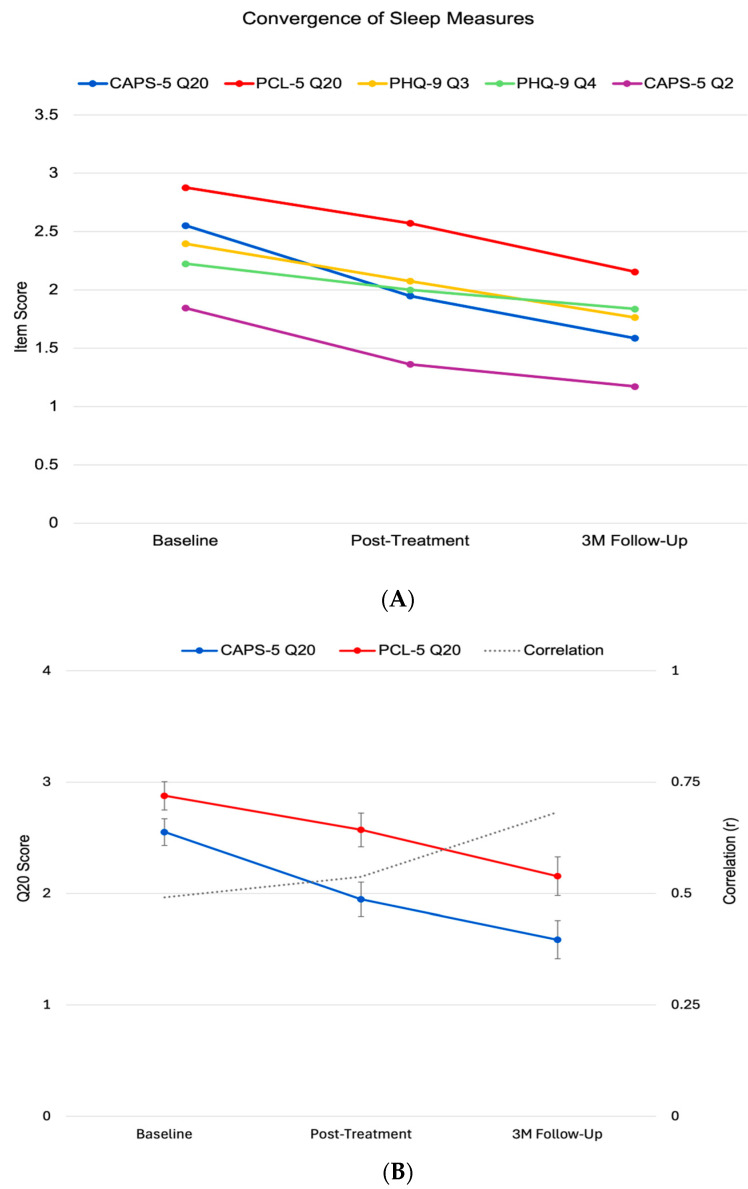
Sleep measure validation and convergence. (**A**) Mean scores (±SE) across timepoints showing parallel trajectories between clinician-rated and self-reported measures. (**B**). Strengthening correlations between CAPS-5 Q20 and PCL-5 Q20 from baseline (r = 0.49) to follow-up (r = 0.68, *p* < 0.01), supporting measurement validity.

**Figure 3 brainsci-15-00297-f003:**
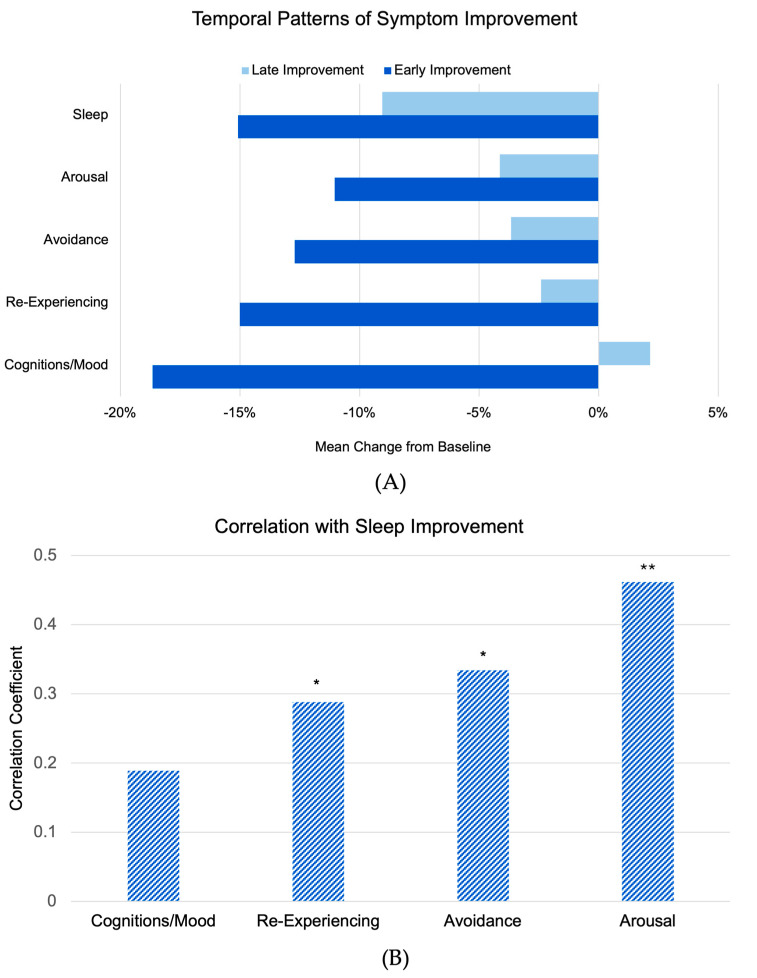
Temporal patterns and correlations of symptom improvement. (**A**) Mean percentage change from baseline showing early improvement (weeks 0–8) and late improvement (weeks 8–12) across symptom clusters. Note the more balanced improvement pattern in sleep symptoms compared to other PTSD symptom domains. (**B**) Correlation coefficients between overall sleep improvement (baseline to 3-month follow-up) and other PTSD symptom clusters, showing strongest association with arousal symptoms (r = 0.46) and weakest with cognitions/mood symptoms (r = 0.19). * *p* < 0.05, ** *p* < 0.0005.

**Figure 4 brainsci-15-00297-f004:**
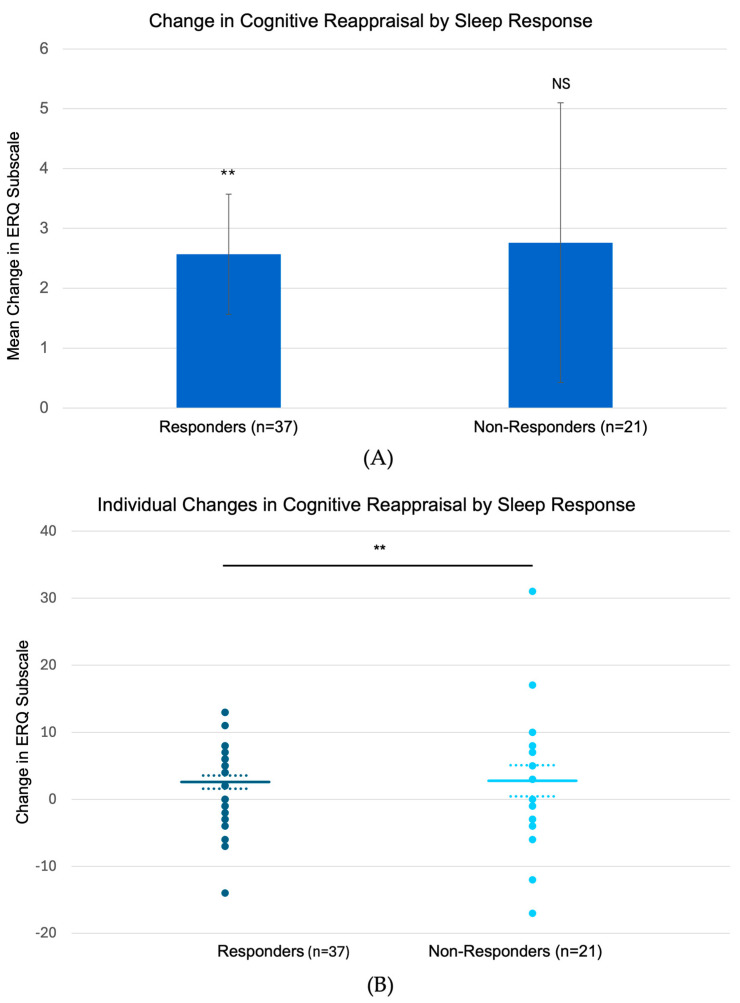
Change in cognitive reappraisal by sleep response status. (**A**) Mean and (**B**) individual change in ERQ CR subscale from baseline to 3-month follow-up, for sleep responders and non-responders. Response defined as ≥1-point reduction in CAPS-5 Q20 at 3M follow-up. Error bars represent standard error of the mean. ** *p* = 0.006.

**Table 1 brainsci-15-00297-t001:** Demographics and clinical characteristics (N = 63).

Characteristic	n (%) or Mean ± SD [Range]
**Demographics**	
Age, years	39.0 ± 10.5 [24.0–63.8]
Sex	
Male	35 (55.6)
Female	28 (44.4)
Education	
≤High school	16 (25.4)
Some college	15 (23.8)
Bachelor’s degree	19 (30.2)
Advanced degree	13 (20.6)
**Trauma Characteristics**	
Context	
Military	30 (47.6)
Civilian	31 (49.2)
Unknown	2 (3.2)
Primary trauma type	
Combat-related	24 (38.1)
Interpersonal violence ^1^	19 (30.2)
Accidents	5 (7.9)
Other	13 (20.6)
Unknown	2 (3.2)
**Clinical Course**	
Time since trauma, years	10.7 ± 5.9 [1.0–20.0]
Time since symptom onset, years	9.4 ± 5.7 [1.0–20.0]
Latency ^2^, years	0.9 ± 2.1 [0–11.0]
**Current Medications**	
SSRI/SNRI	51 (81.0)
Other psychotropic medications ^3^	18 (28.6)

Note: Data presented as n (%) for categorical variables and mean ± standard deviation [range] for continuous variables. SSRI = selective serotonin reuptake inhibitor; SNRI = serotonin-norepinephrine reuptake inhibitor. ^1^ Includes sexual assault/abuse, physical assault/abuse, and domestic violence ^2^ Time between trauma exposure and symptom onset ^3^ Some participants were on multiple medications.

**Table 2 brainsci-15-00297-t002:** Sleep measure convergence.

	Baseline	Post-Treatment	3-Month Follow-Up
**Measure Correlation**			
CAPS-5 Q20 vs. PCL-5 Q20	0.49	0.54	0.68
CAPS-5 Q20 vs. PHQ-9 Q3	0.35	0.6	0.57
CAPS-5 Q20 vs. CAPS-5 Q2	0.32	0.32	0.32
**Mean Scores (±SE)**			
CAPS-5 Q20	2.55 ± 0.12	1.95 ± 0.15	1.59 ± 0.17
PCL-5 Q20	2.87 ± 0.13	2.57 ± 0.15	2.16 ± 0.17
PHQ-9 Q3	2.40 ± 0.11	2.08 ± 0.14	1.76 ± 0.14
CAPS-5 Q2	1.84 ± 0.18	1.36 ± 0.15	1.17 ± 0.16

**Table 3 brainsci-15-00297-t003:** Sleep measure trajectories by response status.

	Responders (n = 37)		Non-Responders (n = 21)	
**Measure**	**Early Change**	**Late Change**	**Early Change**	**Late Change**
PCL-5 Q20	−0.57 ± 0.12 **	−1.14 ± 0.22 ***	0.05 ± 0.29	0.14 ± 0.22
PHQ-9 Q3	−0.65 ± 0.19 ***	−3.9 ± 0.26 ***	−0.29 ± 0.31	0.14 ± 0.25
CAPS-5 Q2	−0.43 ± 0.23 *	−0.46 ± 0.16 **	−0.57 ± 0.27 *	0.05 ± 0.31

Note: Values represent mean ± SEM. * *p* < 0.05, ** *p* < 0.01, *** *p* < 0.001. Early change: baseline to post-treatment; late change: baseline to follow-up.

## Data Availability

Data are available, through reasonable requests to the corresponding author. The data is not publicly available for ethical and commercial reasons.

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
