# Peer review of "Can Amygdala-Derived-EEG-fMRI-Pattern (EFP) Neurofeedback Treat Sleep Disturbances in PTSD?"

_brainsci, 2025, doi:10.3390/brainsci15030297_

Round 1

Reviewer 1 Report

Comments and Suggestions for Authors

The study addresses an important and understudied aspect of PTSD treatment which sleep disturbances and explores an innovative neuromodulation approach. Also, focus on temporal patterns of improvement adds a novel angle to PTSD sleep research. The manuscript presents a well-defined research question and hypotheses that are logically derived from existing literature.

However, there are some caveats which need to be addressed before the manuscript is considered further.

I provide detailed comments below to help the authors improve the manuscript.

  1. The study is a secondary analysis of an open-label, single-arm trial, which limits its ability to establish causal relationships. There is no control group, making it impossible to determine whether observed improvements were due to the intervention or natural symptom fluctuation.
  2. Sleep disturbances are assessed exclusively through subjective self-reported and clinician-administered scales. There are no objective measures such as actigraphy or polysomnography to confirm sleep improvements.
  3. The manuscript suggests that sleep improvements are linked to emotion regulation changes but does not provide neurobiological data to support this claim.
  4. The study only assesses outcomes up to three months post-treatment. Longer follow up is required.
  5. The conclusion states that "Amyg-EFP-NF shows promise for treating PTSD-related sleep disturbances" without acknowledging that this is a preliminary finding from a non-controlled study.
  6. The term “Amygdala-EFP-Neurofeedback” is used inconsistently throughout the manuscript. It should be standardized.
  7. Abstract and the discussion refers to “balanced improvement patterns,” but the explanation is vague. A clearer breakdown of early vs. late changes is needed.
  8. Figure 3 shows late/early improvement ratios, but it is not clearly explained how these were calculated.
  9. Figure 4: individual data points should be presented in the graph.
  10. The manuscript does not explore why 36.21% of participants did not improve.
  11. Were there baseline differences between responders and non-responders? Did certain clinical or demographic factors predict treatment response?
  12. A subgroup analysis could reveal whether initial PTSD severity, medication status, or trauma type influenced response rates.
  13. The discussion does not adequately compare Amyg-EFP-NF with other evidence-based PTSD sleep treatments (e.g., Cognitive Behavioral Therapy for Insomnia [CBT-I], prazosin for nightmares).
  14. The study states that sleep improved more gradually than other PTSD symptoms, but does not clearly quantify how much improvement occurred in early vs. late phases.
  15. The study reports that sleep improvement correlated moderately with PTSD symptom reduction (r = 0.484, p < 0.001). However, it does not examine whether improvement in sleep correlated with specific PTSD symptom clusters (e.g., hyperarousal, re-experiencing).

Author Response

Responses to reviewer

We sincerely thank the reviewer for their thoughtful and constructive feedback on our manuscript. We appreciate the recognition of our work's relevance and innovation in the field. We have addressed each comment as detailed below:

Introduction

Comment: This section is very clear, the ideas well organized, and concisely presented, and a well formulated problem. My comments regarding to this section are related to the evidence and theory for the selected treatment. The authors only mention about the Amygdala-derived-EEG-fMRI-Pattern, but no theory, technical, and neural mechanisms are described in detail to provide the readers of a reliable reference regarding the intervention for this study. Also if Prism™ procedures are reported elsewhere would be interesting to briefly describe them in this section, for example after lines 67-69 when citing reference 14 (Amygdala-Derived-EEG-FMRI-Pattern Neurofeedback for the Treatment of Chronic Post-Traumatic Stress Disorder).

Response: We agree with the reviewer and have expanded the introduction to include a more detailed explanation of the Amyg-EFP neurofeedback theoretical framework, technical aspects, and neural mechanisms. We have added the following paragraph after line 69:

"The Amyg-EFP neurofeedback system represents an innovative approach to targeted neuromodulation. This technology integrates simultaneous EEG and fMRI recordings through a machine learning algorithm that identifies specific EEG patterns correlating with amygdala activity (Keynan et al., 2016). Unlike traditional neurofeedback methods that have low spatial resolution and cannot capture neural activity from deeper brain regions, the Amyg-EFP approach enables indirect access to amygdala functioning using only scalp EEG. The Prism™ system operationalizes this technology by providing real-time feedback based on these amygdala-derived EEG patterns through an audio-visual interface. This allows patients to modulate their emotional state implicitly, without requiring explicit strategy instructions, potentially facilitating downregulation of the amygdala hyperactivity associated with both PTSD symptoms and sleep disturbances (Voigt et al., 2024; Fruchter et al., 2024)."

Materials and Methods

Comment: 2.1 Study design and participants. In this section, the authors report <5% of missing data, were these cases excluded from the sample? Or included as cases? Please specify, and justify if retained in the sample even with the missing data. Line 94, please specify what does “Five sites” mean, is it a proper noun?

Is there any ethical procedures approval? The authors refer to an informed consent, but there is no information provided about an ethical approval for this study.

Response: We thank the reviewer for highlighting these important points. We have revised the manuscript as follows:

  1. Missing data: We have added the following clarification in section 2.1: "Missing data were minimal (<5%) due to high retention rates. Complete case analysis was used for our primary analyses, as the small amount of missing data was determined to be missing completely at random based on the distribution of missing values."
  2. "Five sites": We have clarified this in section 2.1: "Recruitment occurred through clinical referrals and advertisements at five clinical centers: four in Israel (Rambam Medical Center, Sheba Medical Center, Be'er Ya'akov Mental Health Center, Barzilai Medical Center) and one in the United States (New York University Langone Health)."
  3. Ethical approval: We have added the following statement to section 2.1: "The study was approved by the ethics committees at each participating clinical site and the trial was registered on ClinicalTrials.gov (NCT04891614)."

Comment: Another important issue in the instruments section is that not all of the scales explain the punctuation, for example, it is reported for 2.1.1 Sleep Assessment, but not for 2.2.2 Emotion regulation, and 2.2.3 PTSD. I suggest the authors to briefly explain how are these scales interpreted, or if only means and standard deviations were obtained for this research purposes. This is relevant, considering that in the results section, scores are presented in continuous variables, but it is not clear how they were processed.

Response: We appreciate this observation and have added the following information to sections 2.2.2 and 2.2.3:

For section 2.2.2 (Emotion Regulation Assessment): "The ERQ uses a 7-point Likert scale (1 = strongly disagree to 7 = strongly agree). Scores on the CR subscale range from 6-42, scores on the ES subscale range from 4-28, with higher scores indicating greater use of each strategy."

For section 2.2.3 (PTSD Symptom Assessment): "The CAPS-5 uses a 5-point severity scale (0 = absent to 4 = extreme/incapacitating) for each symptom. The total CAPS-5 score ranges from 0-80, with scores categorized as: minimal (0-15), mild (16-25), moderate (26-45), severe (46-65), and extreme (66-80)."

Comment: 2.4 Statistical Analysis, in this section, the authors refer to the use of t-tests, but there is no reference for Chi-Square which is also reported later, as well as correlation tests, reported in section 3.2 Measurement Validation and Covariance.

Response: We have expanded section 2.4 (Statistical Analysis) to include all statistical methods used:

"Changes in emotion regulation strategies were compared between sleep responders and non-responders using independent t-tests. Chi-square tests were used to compare categorical variables, including treatment response rates between trauma types. Pearson correlation coefficients were calculated to examine relationships between sleep measures and between sleep improvement and overall PTSD symptom reduction. To examine temporal patterns, we defined early improvement as change from baseline to 8 weeks, and late improvement as change from 8 weeks to 3 months..."

Discussion

Comment: The discussion is a well centered section of the manuscript, explaining a good framework of the manuscript presenting a symptoms and treatment analysis, but it would improve if the authors consider including neurofeedback findings from other studies, for being a neurofeedback centered research, in general, introduction and discussion lacks of literature addressing findings from other neurofeedback studies.

A minor comment for this section for authors consideration is reserving statistical data only for the results section.

Response: We agree with these suggestions and have made the following changes:

  1. We have expanded the discussion to include more neurofeedback literature by adding the following paragraph in section 4:

"Our findings extend previous neurofeedback research in sleep and emotional disorders. The Amyg-EFP approach differs from traditional neurofeedback by providing feedback on activity patterns specifically linked to the amygdala, potentially offering more targeted modulation of emotional regulation circuits. Previous research has shown that amygdala-targeting neurofeedback facilitates learned modulation (downregulation) of the amygdala (Goldway et al., 2022), with preliminary evidence demonstrating significant reductions in PTSD symptoms following such interventions (Zotev et al., 2018; Nicholson et al., 2020)."

  1. We have removed statistical data from the discussion section and ensured all statistical findings are properly presented only in the results section.

We hope these revisions adequately address the reviewer's comments and strengthen our manuscript. We are grateful for the valuable feedback that has helped improve the clarity and quality of our work.

Reviewer 2 Report

Comments and Suggestions for Authors

I thank the Editor and the authors for the opportunity to review the manuscript entitled: Can Amygdala-derived-EEG-fMRI-Pattern(EFP)-Neurofeedback treat sleep disturbances in PTSD? Submitted to MDPI BrainSci_3518355.

The problem approached by the researchers is of great relevance, and explores objective and innovative variables in the field.

Introduction

This section is very clear, the ideas well organized, and concisely presented, and a well formulated problem. My comments regarding to this section are related to the evidence and theory for the selected treatment. The authors only mention about the Amygdala-derived-EEG-fMRI-Pattern, but no theory, technical, and neural mechanisms are described in detail to provide the readers of a reliable reference regarding the intervention for this study. Also if Prism™ procedures are reported elsewhere would be interesting to briefly describe them in this section, for example after lines 67-69 when citing reference 14 (Amygdala-Derived-EEG-FMRI-Pattern Neurofeedback for the Treatment of Chronic Post-Traumatic Stress Disorder).

Materials and methods

2.1 Study design and participants. In this section, the authors report <5% of missing data, were these cases excluded from the sample? Or included as cases? Please specify, and justify if retained in the sample even with the missing data. Line 94, please specify what does “Five sites” mean, is it a proper noun?

Is there any ethical procedures approval? The authors refer to an informed consent, but there is no information provided about an ethical approval for this study.

Another important issue in the instruments section is that not all of the scales explain the punctuation, for example, it is reported for 2.1.1 Sleep Assessment, but not for 2.2.2 Emotion regulation, and 2.2.3 PTSD. I suggest the authors to briefly explain how are these scales interpreted, or if only means and standard deviations were obtained for this research purposes. This is relevant, considering that in the results section, scores are presented in continuous variables, but it is not clear how they were processed.

2.4 Statistical Analysis, in this section, the authors refer to the use of t-tests, but there is no reference for Chi-Square which is also reported later, as well as correlation tests, reported in section 3.2 Measurement Validation and Covariance.

Discussion

The discussion is a well centered section of the manuscript, explaining a good framework of the manuscript presenting a symptoms and treatment analysis, but it would improve if the authors consider including neurofeedback findings from other studies, for being a neurofeedback centered research, in general, introduction and discussion lacks of literature addressing findings from other neurofeedback studies.

A minor comment for this section for authors consideration is reserving statistical data only for the results section.

Author Response

Responses to reviewer

We thank the reviewer for their thorough evaluation and constructive feedback on our manuscript. We appreciate the recognition of our study's importance in addressing an understudied aspect of PTSD treatment. We have carefully considered all comments and have addressed them as detailed below:

Study Design Limitations

Comments 1, 5: The study is a secondary analysis of an open-label, single-arm trial, which limits its ability to establish causal relationships. There is no control group, making it impossible to determine whether observed improvements were due to the intervention or natural symptom fluctuation.

Response: We fully agree with these important limitations. We have revised our manuscript to more explicitly acknowledge these limitations throughout:

  1. We have modified the abstract conclusion to read: "Amyg-EFP-NF shows preliminary promise for treating PTSD-related sleep disturbances in this exploratory, non-controlled study. Our exploratory analyses suggest distinct temporal patterns of sleep improvement and potential associations with enhanced cognitive reappraisal capacity that warrant rigorous investigation in future randomized controlled trials."
  2. We have added a new paragraph at the beginning of the limitations section:

"Several important limitations must be considered when interpreting our findings. First and foremost, the single-arm design without a control condition precludes definitive conclusions about treatment efficacy, as improvements could reflect natural symptom fluctuation, regression to the mean, or non-specific treatment effects. While the blinding of assessment timepoints during clinician ratings helps mitigate some potential biases, the exploratory nature of this analysis means our findings should be considered hypothesis-generating rather than confirmatory. The results presented here provide preliminary evidence that justifies further investigation in properly controlled trials."

Assessment Methods

Comment 2: Sleep disturbances are assessed exclusively through subjective self-reported and clinician-administered scales. There are no objective measures such as actigraphy or polysomnography to confirm sleep improvements.

Response: We agree that objective sleep measures would strengthen our findings. We have expanded the limitations section to address this more thoroughly:

"Additionally, while our sleep measures showed good convergence and are validated assessments in PTSD populations, the absence of objective sleep measurements (e.g., actigraphy or polysomnography) limits our ability to assess specific sleep architecture changes. Future studies should incorporate these objective measures to better characterize the nature of sleep improvements and potential mechanisms. These objective measures could help differentiate effects on different sleep parameters (e.g., sleep onset latency, sleep efficiency, REM architecture) and provide insights into potential relationships between objectively measured sleep quality and subjective improvements."

Neurobiological Mechanisms

Comment 3: The manuscript suggests that sleep improvements are linked to emotion regulation changes but does not provide neurobiological data to support this claim.

Response: This is a valid point. We have revised the discussion to clarify the preliminary nature of these findings and acknowledge this limitation:

"While our findings suggest a potential link between improved emotional regulation and sleep outcomes, we acknowledge the absence of direct neurobiological data to establish this mechanistic relationship. Our observed associations between cognitive reappraisal and sleep improvement provide intriguing but preliminary support for emotional regulation as a potential recovery mechanism. Future studies should incorporate neuroimaging measures before and after treatment to directly assess changes in prefrontal-amygdala connectivity and their relationship to both emotional regulation capacity and sleep quality improvements."

Follow-up Duration

Comment 4: The study only assesses outcomes up to three months post-treatment. Longer follow up is required.

Response: We agree that longer follow-up would be valuable. We have added the following to the limitations section:

"Our study assessed outcomes only up to three months post-treatment, which may not capture the full trajectory of sleep improvement or potential relapse patterns. Longer follow-up periods (e.g., 6-12 months) in future studies would provide important information about the durability of treatment effects and could help identify factors associated with sustained improvement versus symptom recurrence."

Terminology Consistency

Comment 6: The term “Amygdala-EFP-Neurofeedback” is used inconsistently throughout the manuscript. It should be standardized.

Response: We appreciate this observation and have standardized the terminology throughout the manuscript. We now consistently use "Amyg-EFP-NF" after first defining the full term "Amygdala-derived-EEG-fMRI-Pattern Neurofeedback (Amyg-EFP-NF)" in the introduction.

Clarification of Improvement Patterns

Comments 7, 8, 14: Abstract and the discussion refers to “balanced improvement patterns,” but the explanation is vague. A clearer breakdown of early vs. late changes is needed. Figure 3 shows late/early improvement ratios, but it is not clearly explained how these were calculated. The study states that sleep improved more gradually than other PTSD symptoms, but does not clearly quantify how much improvement occurred in early vs. late phases.

Response: We have addressed these points with the following changes:

  1. We have added a clearer explanation in the methods section:

"To examine temporal patterns, we defined early improvement as the change from baseline to post-treatment (8 weeks), and late improvement as the change from post-treatment to 3-month follow-up. For each symptom cluster, we calculated percentage improvement in each phase by dividing the absolute change by the maximum possible score for that cluster, allowing for standardized comparison across clusters. The late/early improvement ratio was calculated by dividing the percentage improvement in the late phase by the percentage improvement in the early phase. A ratio of 1.0 would indicate equal improvement in both phases, while values below 1.0 indicate greater early improvement, and values above 1.0 indicate greater late improvement."

  1. We have added specific quantification in the results section addressing the temporal improvement patterns for sleep and other symptom clusters and revised the text of “Temporal Patterns of Improvement”:

“Sleep improvement demonstrated a distinct temporal pattern compared to other symptom clusters, with late/early improvement ratios revealing notable differences between clusters (Figure 3). Sleep improvement showed a balanced pattern across phases (-15.1% early phase, -9.1% late phase) with a late/early ratio of 0.60, contrasting with other symptom clusters that showed predominantly early improvement followed by minimal late-phase change: Arousal (-11.0% early, -4.1% late, ratio 0.38), Cognitions/Mood (-18.7% early, +2.2% late [small worsening], ratio -0.12), Avoidance (-12.7% early, -3.7% late, ratio 0.29), and Re-experiencing (-15.0% early, -2.4% late, ratio 0.16).

When examining total symptom reduction across the study period, overall improvement was greatest for sleep (-38.5%, SEM=7%) followed by Re-experiencing (-30.7%, SEM=6%), Cognitions/Mood (-28.8%, SEM=4%), Arousal (-24.0%, SEM=7%), and Avoidance (-21.7%, SEM=7%). Comparative statistical analysis revealed highly significant differences between the late/early improvement ratio for sleep and that of Cognitions/Mood (p=0.0001), significant differences with Re-experiencing (p=0.0094), a marginally significant difference with Avoidance (p=0.0541), and no significant difference with Arousal (p=0.1383). While other symptom clusters showed predominantly early improvement followed by minimal late-phase change or slight regression, sleep disturbances exhibited a more nuanced recovery pattern with substantial improvement continuing into the late phase.”

Data Visualization

Comment 9: Figure 4: individual data points should be presented in the graph.

Response: We have revised Figure 4 to include another panel showing individual data points alongside the mean values and error bars, providing a more complete visualization of the data distribution and individual variability in cognitive reappraisal changes.

Non-responder Analysis

Comments 10, 11, 12: The manuscript does not explore why 36.21% of participants did not improve. Were there baseline differences between responders and non-responders? Did certain clinical or demographic factors predict treatment response? A subgroup analysis could reveal whether initial PTSD severity, medication status, or trauma type influenced response rates.

Response: We thank the reviewer for these excellent suggestions. We have conducted additional analyses and added a new section titled “Predictors of Treatment Response” that examines differences between sleep responders and non-responders:

"We examined baseline characteristics of sleep responders (n=37) versus non-responders (n=21) to identify potential predictors of treatment response. There were no significant baseline differences in demographic factors including age (responders: 38.95±10.42 years, non-responders: 38.33±10.58 years, p=0.83), gender distribution (responders: 46% female, non-responders: 48% female, p=0.90), or trauma type (responders: 51% non-military, non-responders: 48% non-military, p=0.85).

Clinical characteristics including time from symptom onset (responders: 8.35±0.89 years, non-responders: 10.91±1.38 years, p=0.11), baseline PTSD severity (CAPS-5 total: responders 43.19±2.00, non-responders 42.71±2.35, p=0.87), and symptom cluster severity (all p>0.30) also did not significantly differ between groups. Baseline emotion regulation capacity, as measured by the ERQ cognitive reappraisal (responders: 24.35±1.48, non-responders: 25.00±1.93, p=0.78) and expressive suppression (responders: 15.43±0.97, non-responders: 15.29±1.41, p=0.93) subscales, was similar in both groups.

However, baseline sleep disturbance severity (CAPS-5 Item 20) was marginally higher in responders (2.70±0.16) compared to non-responders (2.29±0.18, p=0.08), suggesting that patients with more severe sleep problems may have greater potential for improvement. This finding aligns with our categorical analysis of response rates by baseline severity, which showed increasing response rates with higher baseline sleep disturbance severity.

Further analysis of response rates by baseline sleep severity revealed that among participants with mild baseline sleep disturbance (CAPS-5 Item 20=1, n=8), 50% responded at 3-month follow-up, compared to 61% of those with moderate severity (Item 20=2, n=18), 58% with severe disturbance (Item 20=3, n=24), and 100% of those with extreme severity (Item 20=4, n=8). Similarly, higher overall PTSD symptom severity predicted better sleep outcomes, with response rates of 58.6% for moderate PTSD (baseline CAPS-5 total: 21-40, n=29), 63.0% for severe PTSD (CAPS-5 total: 41-60, n=27), and 100% for the most severe cases (CAPS-5 total: 61-80, n=3) at 3-month follow-up.

These findings suggest that Amyg-EFP-NF may be particularly beneficial for patients with more severe sleep disturbances, possibly due to the intervention's targeted impact on limbic hyperarousal mechanisms that may be more pronounced in severe cases.”

Comparison with Other Treatments

Comment 13: The discussion does not adequately compare Amyg-EFP-NF with other evidence-based PTSD sleep treatments (e.g., Cognitive Behavioral Therapy for Insomnia [CBT-I], prazosin for nightmares).

Response: We agree and have added a new paragraph to the discussion that compares our findings with existing PTSD sleep treatments:

"Our findings should be considered in the context of existing evidence-based PTSD sleep treatments. Cognitive Behavioral Therapy for Insomnia (CBT-I) is well-established as the first-line treatment for insomnia in PTSD, with meta-analyses reporting moderate to large effects (Ho et al., 2016). The effect size observed in our study (d = 0.54 for the full sample) is comparable to those reported for sleep treatments in PTSD populations. Recent network meta-analysis by Huang et al. (2024) identified CBT-I as having the highest efficacy for improving sleep quality (SMD = -5.61, 95%CI: -8.82 to -2.40 compared to placebo), while pharmacological treatments showed varying degrees of effectiveness. For nightmare reduction, this same analysis found prazosin (SMD = -1.20, 95%CI: -1.72 to -0.67) and imagery rehearsal therapy (IRT) (SMD = -0.65, 95%CI: -1.00 to -0.31) to be most effective. Our observed effects on trauma-related dreams, particularly the sustained improvement in responders, suggest potential efficacy in this domain as well, though direct comparative studies are needed. The Amyg-EFP-NF approach differs mechanistically by targeting emotional regulation circuits rather than directly modifying sleep behaviors (as in CBT-I) or pharmacologically altering adrenergic activity (as with prazosin). This mechanistic difference may offer advantages for patients who have not responded to conventional approaches or prefer non-pharmacological treatments. Additionally, our finding that patients with more severe baseline sleep disturbances showed better responses contrasts with evidence suggesting that traditional approaches may be less effective for severe cases, particularly those with short sleep duration (Miller et al., 2020).”

Correlations with Symptom Clusters

Comment 15: The study reports that sleep improvement correlated moderately with PTSD symptom reduction (r = 0.484, p < 0.001). However, it does not examine whether improvement in sleep correlated with specific PTSD symptom clusters (e.g., hyperarousal, re-experiencing).

Response: We thank the reviewer for this suggestion and have conducted additional analyses examining correlations between sleep improvement and changes in specific PTSD symptom clusters. The following text was added to the results: “We examined correlations between sleep improvement and changes in specific PTSD symptom clusters (Figure 3B). Sleep improvement (change in CAPS-5 Item 20) showed strongest correlation with improvement in the arousal cluster (excluding the sleep item) (r = 0.46, p < 0.001), followed by avoidance symptoms (r = 0.33, p = 0.012) and re-experiencing symptoms (r = 0.29, p = 0.027). The correlation with negative alterations in cognition and mood was weaker and did not reach statistical significance (r = 0.19, p = 0.153). The stronger association with other arousal symptoms suggests potential shared regulatory mechanisms, consistent with neurobiological models of hyperarousal affecting both daytime symptoms and sleep disruption.”

We believe these revisions comprehensively address the reviewer's comments and significantly strengthen our manuscript. We greatly appreciate the detailed feedback that has helped improve the clarity, rigor, and impact of our work.

Round 2

Reviewer 1 Report

Comments and Suggestions for Authors

The authors have adequately addressed my comments.